# Anatomy-Grounded Weakly Supervised Prompt Tuning for Chest X-ray Latent Diffusion Models

**Konstantinos Vilouras**[1] (ID)           KONSTANTINOS.VILOURAS@ED.AC.UK
[1] *School of Engineering, University of Edinburgh*

**Ilias Stogiannidis**[1] (ID)              I.STOGIANNIDIS@ED.AC.UK
**Junyu Yan**[1] (ID)                 JUNYU.YAN@ED.AC.UK
**Alison Q. Smithard**[1,2] (ID)      ALISON.SMITHARD@MRE.MEDICAL.CANON
[2] *Canon Medical Research Europe Ltd.*

**Sotirios A. Tsaftaris**[1] (ID)            S.TSAFTARIS@ED.AC.UK

**Editors:** Accepted for publication at MIDL 2026

## Abstract

Latent Diffusion Models have shown remarkable results in text-guided image synthesis in recent years. In the domain of natural (RGB) images, recent works have shown that such models can be adapted to various vision-language downstream tasks with little to no supervision involved. On the contrary, text-to-image Latent Diffusion Models remain relatively underexplored in the field of medical imaging, primarily due to limited data availability (e.g., due to privacy concerns). In this work, focusing on the chest X-ray modality, we first demonstrate that a standard text-conditioned Latent Diffusion Model has not learned to align clinically relevant information in free-text radiology reports with the corresponding areas of the given scan. Then, to alleviate this issue, we propose a fine-tuning framework to improve multi-modal alignment in a pre-trained model such that it can be efficiently repurposed for downstream tasks such as phrase grounding. Our method sets a new state-of-the-art on a standard benchmark dataset (MS-CXR), while also exhibiting robust performance on out-of-distribution data (VinDr-CXR). We further validate our approach through a pilot qualitative study and an experiment on grounded disease classification. Our code will be made publicly available at `https://github.com/vios-s`.

**Keywords:** Chest X-rays, Text-to-Image Latent Diffusion Models, Phrase Grounding

## 1. Introduction

Latent Diffusion Models (LDMs) ([Rombach et al., 2022](#)) form a class of powerful text-to-image generators that achieve state-of-the-art (SOTA) performance in conditional image synthesis. Recently, the research community has shown increasing interest in the applicability of LDMs to various downstream tasks such as image editing ([Mokady et al., 2023](#)), semantic correspondence ([Luo et al., 2023](#)), depth estimation ([Ke et al., 2024](#)) and keypoint detection ([Hedlin et al., 2024](#)), with minimal supervision. In the context of biomedical vision-language processing (VLP), and focusing on the chest X-ray (CXR) modality in particular, LDMs have been repurposed to stress test task-specific models ([Pérez-García et al., 2024](#)), improve their robustness to distribution shifts via synthetic data augmentation ([Ktena et al., 2024](#)), or directly as classifiers ([Favero et al., 2025](#)).

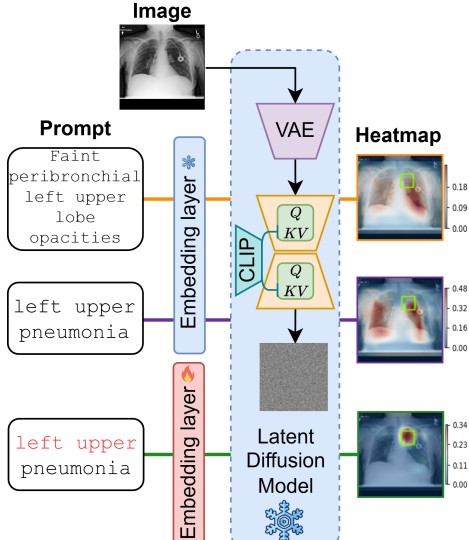

Figure 1: Cross-attention leakage in a pre-trained LDM. Using a description of findings extracted from the radiology report as a prompt (orange line path), we observe that the resulting cross-attention activations (averaged across selected layers and timesteps) are diffused over the image area. Also, following a simpler "{location} {pathology}" prompt format (purple line path) clearly does not improve the activations. Instead, our proposed weakly supervised fine-tuning method (green line path) yields more localized cross-attention activations with respect to the anatomical area mentioned in the prompt.

However, as shown in Figure 1, a closer inspection of the pre-trained LDM's cross-attention layers reveals a high level of *attention leakage*, i.e., the effect of a model associating tokens with unrelated image regions. Note that this finding is consistent with prior work (McInerney et al., 2022) that showed similar behavior in self-supervised models (trained with a multi-modal contrastive learning objective) and proposed few-shot fine-tuning with ground truth pathology bounding boxes as a solution. Furthermore, one could argue that this issue can be attributed to the high complexity of sentences extracted from unstructured radiology reports; yet, as shown in Figure 1, establishing a standardized, simpler prompt format (referred to as "{location} {pathology}") does not alleviate this problem.

The ability to accurately ground findings is crucial for clinical deployment. High-quality heatmaps can (i) facilitate clinician communication by explicitly highlighting visual evidence supporting diagnostic conclusions during case handoffs, or (ii) serve as interactive teaching tools for medical trainees on platforms such as Radiopaedia[1].

In this study, we draw inspiration from techniques that rely on guidance from another modality to improve image-text alignment. For example, there exist works (Ma et al., 2024) that incorporate eye tracking data (collected while a radiologist was examining a CXR scan) during the vision-language pre-training stage. Instead, to avoid the need to collect inputs from another modality, we derive coarse supervision signals directly from free-text reports

---

1. https://radiopaedia.org/

with the help of a pre-trained clinical entity recognition model and a small set of anatomical location annotations. Moreover, we propose a fine-tuning method that refines the model's multi-modal alignment in a data- and parameter-efficient manner by simply updating the anatomy token embeddings.

Overall, our **contributions** are the following:

- We propose a novel approach to improving image-text alignment in biomedical VLP scenarios by extracting a supervision signal for pathology localisation directly from unstructured radiology reports. To this end, we combine a clinical entity recognition model with annotations of various anatomical regions commonly depicted in CXRs.

- We develop an efficient fine-tuning framework to steer the pre-trained LDM's cross-attention activations towards the anatomical area specified in text.

- We evaluate our proposed approach on an established phrase grounding benchmark dataset (MS-CXR), as well as an out-of-distribution (OOD) dataset (VinDr-CXR) with ground truth bounding box annotations and synthetic prompts. In both cases, we show superior performance compared to previous SOTA. We also provide empirical evidence of our method's utility through a pilot qualitative study and an experiment on grounded disease classification.

## 2. Related Work

This section situates our work within three core research areas: text-to-image latent diffusion models for chest X-rays, existing methodologies for the task of medical phrase grounding, and the application of RadGraph-based entity extraction in various downstream tasks.

### 2.1. Chest X-ray Latent Diffusion models

This section provides an overview of existing works on text-to-image Latent Diffusion Models trained on chest X-ray data. For a comprehensive review covering other imaging modalities and applications, please refer to Kazerouni et al., 2023. In this context, Chambon et al., 2022 showed that fine-tuning only the U-Net module of Stable Diffusion is enough to adapt to the X-ray modality. Moreover, textual inversion (i.e., defining new tokens and learning their embeddings while keeping the text encoder frozen) can also be used to fine-tune Stable Diffusion in a few-shot manner (De Wilde et al., 2023). Weber et al., 2023 extend the LDM pipeline with a super-resolution diffusion process in the decoded image space and train their model on a large collection of publicly available chest X-ray datasets ($\sim$ 650k images). Gu et al., 2023 consider the task of counterfactual image generation by first training a LDM on (image, report) pairs and then on (prior image, progression description, current image) triplets, where GPT-4 provides descriptions of disease progression. More recently, Kumar et al., 2025 fine-tuned Stable Diffusion on the CheXpert dataset (Irvin et al., 2019) for text-guided counterfactual image generation, whereas Huang et al., 2024 proposed a custom lightweight diffusion model architecture that was trained on the MIMIC-CXR database (Johnson et al., 2019).

## 2.2. Phrase grounding

The task of linking entities mentioned in text to the corresponding image regions is commonly referred to as phrase grounding. In the domain of natural (RGB) images, there exist large-scale image-text datasets provided with bounding box annotations that enable end-to-end training (see Figure 1 in Dai et al., 2024 for a visual overview of existing approaches). On the contrary, publicly available CXR datasets that contain images, ground truth bounding boxes, and paired radiology reports remain exceedingly rare (with the exception of the small-scale MS-CXR dataset (Boecking et al., 2022) which is only used for evaluation). Moreover, the provided bounding box annotations are typically limited, thus pathology (or anatomy) detectors trained on such datasets do not transfer well on out-of-distribution data (e.g., scans from different hospitals). As a result, while there exist supervised approaches focused on direct bounding box regression (Chen et al., 2023; Zhang et al., 2025), most robust methods developed for medical phrase grounding rely on modality-specific encoders that align image and text features via late fusion (Huang et al., 2021; Boecking et al., 2022; Bannur et al., 2023). There also exist recent works that evaluate pre-trained LDMs on medical phrase grounding in zero-shot (Dombrowski et al., 2024; Vilouras et al., 2024; Nützel et al., 2025).

## 2.3. Integrating RadGraph into downstream tasks

Prior works have used RadGraph-1.0 (Jain et al., 2021) annotations to evaluate or even further improve the performance of task-specific models. For example, Yu et al., 2023 proposed the RadGraph F1 evaluation metric for radiology report generation, while Delbrouck et al., 2022 showed improvements on this task by optimizing RadGraph-based rewards with reinforcement learning. In the more general context of biomedical VLP, Yu et al., 2022 derive classification labels from RadGraph's "observation-located-at" relations and train two separate anatomy and pathology classifiers with a binary cross-entropy loss. More recently, Varma et al., 2023 and Wu et al., 2023 showed that RadGraph annotations can further boost the performance of image-text contrastive learning methods.

## 3. Method

We propose a weakly supervised fine-tuning method for LDMs to improve pathology localisation using anatomical references provided in free-form radiology reports. To this end, we first design a pipeline based on a clinical entity recognition model (RadGraph-XL) and a small set of annotations covering different anatomical regions, to derive a weak 2D supervision signal following a Gaussian distribution (Section 3.1). Then, we formulate an optimization objective to update the anatomy token embeddings using a dynamically-generated target, obtained by linearly mixing the LDM's cross-attention activations with the Gaussian. Moreover, to encourage compositional interactions between anatomy and pathology tokens within prompts, we incorporate a loss term that penalizes overlapping cross-attention activations across tokens (Section 3.2). The key insight of our approach is that refining token-level representations, while keeping the feature extraction components of the LDM (i.e., the VAE image encoder, the CLIP text encoder, and the denoising U-Net) frozen, can effectively improve performance on a downstream image-text alignment task.

**1. LUT design (based on Chest Imagenome annotations)**

a. List of anatomical locations    b. Bounding box annotations in $[x, y, w, h]$ format    c. Box-level statistics    d. Location-level statistics

left lung

right lung

cardiac silhouette

$$\mu_x = \frac{2x + w}{2}$$

$$\mu_y = \frac{2y + h}{2}$$

$$\sigma_x = \frac{w}{6}$$

$$\sigma_y = \frac{h}{6}$$

$\mathbb{E}[\cdot]$

| LUT | |
|---|---|
| **Location** | **Gaussian parameters** |
| left lung | $\mu_x, \mu_y, \sigma_x, \sigma_y$ |
| right lung | $\mu_x, \mu_y, \sigma_x, \sigma_y$ |
| cardiac silhouette | $\mu_x, \mu_y, \sigma_x, \sigma_y$ |

**2. Data curation**

**Prompt**

Mild bibasilar atelectasis in the setting of low lung volumes.

**BiomedVLP-CXR-BERT (RadGraph-XL)**

"ANAT-DP"

bibasilar

+ GT label

**Standardized prompt**

bibasilar atelectasis

**Image**

$\mathcal{N}(\mu; \sigma)$

**LUT**

Figure 2: Overview of our proposed data curation process. (*top*) Using Chest Imagenome (Wu et al., 2021) annotations, we design a lookup table (LUT) that summarizes the first-order statistics of each anatomical area. The inferred parameters are used to define a Gaussian distribution per location. (*bottom*) Raw sentences are processed by the RadGraph-XL model (Delbrouck et al., 2024) to identify location tokens. In turn, those tokens are used to retrieve the corresponding Gaussian(s), and to construct prompts in the format "{location} {pathology}".

### 3.1. Data curation

We now describe our proposed approach for curating a dataset that contains triplets of images (CXRs), prompts in a standardized format ("{location} {pathology}"), and a coarse supervision signal that approximately highlights the specified anatomical area on the input image in the form of a Gaussian. An overview of our approach is shown in Figure 2. Additional implementation details for the data curation process are in Appendix D.

Since radiology reports contain free-form text, we use the pre-trained RadGraph-XL (Delbrouck et al., 2024), which is a BERT model designed to extract clinical entities and their relations, to identify location modifiers within individual sentences. Such terms are labeled as *Anatomy definitely present*, or "ANAT-DP" in short, by the model. Then, we apply a rule-based post-processing step (detailed in Appendix D) to merge similar terms that refer to specific anatomical regions. As a result, our curated dataset used for fine-tuning consists of 6,480 samples spanning 27 anatomical locations and 8 pathologies in total. A detailed description of the dataset's statistics is presented in the Appendix in Figure 10.

Moreover, since our ultimate goal is to steer the cross-attention activations of the pre-trained LDM towards the anatomical locations specified in text, we need to translate the "ANAT-DP" entities extracted from RadGraph-XL into a spatial signal corresponding to

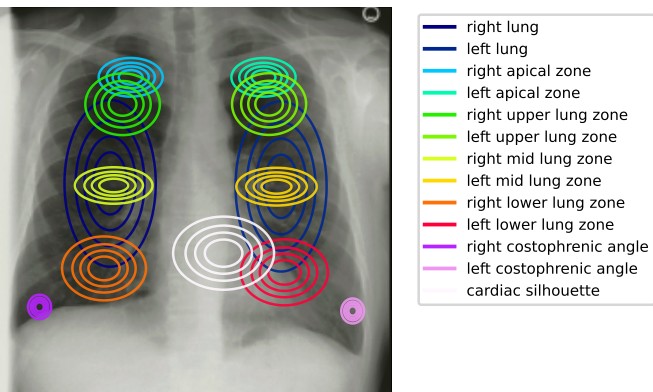

Figure 3: Mapping from anatomical locations to 2D Gaussians based on Chest Imagenome (Wu et al., 2021) gold standard annotations. For clarity, we overlay the Gaussians on top of a randomly selected CXR from the same dataset. Note that we use these targets during our proposed fine-tuning method.

the input image. To this end, we design a lookup table (LUT) that maps location tokens into 2D Gaussian(s) with pre-defined parameters $\mu$ and $\sigma$. Those parameters are calculated from the gold standard subset of the Chest Imagenome (Wu et al., 2021) dataset that contains 1,000 images with bounding box annotations per anatomical location. Note that the overlap between Chest Imagenome and our fine-tuning set is small (28 images out of 6,480 used for fine-tuning in total). A schematic overview of the LUT design stage is presented in Figure 2, while a detailed discussion of this mapping process is deferred to Appendix D. We also present the final set of 2D Gaussians derived from Chest Imagenome in Figure 3.

### 3.2. Prompt tuning

Given the curated dataset described in the previous section, our goal now is to fine-tune the token embeddings such that cross-attention activations are restricted to the location specified in the input prompt. To this end, we initialize a codebook of size (46, 1024) based on the pre-trained CLIP embeddings used during the initial LDM training stage, where 46 is the total number of (sub-)tokens including location terms, pathology terms and special tokens ($\langle\text{BoS}\rangle$, $\langle\text{EoS}\rangle$, $\langle\text{pad}\rangle$, and the $\langle\text{and}\rangle$ token which is used for binding), and 1024 is the feature dimensionality. Note that the pathology and special token embeddings remain frozen during fine-tuning. Then, following the reverse diffusion process (mapping from noise to image latents), we extract the intermediate cross-attention activations from 30 timesteps (in range [30, 60]) and 4 layers (i.e., 1 from the U-Net's bottleneck and the first 3 from the U-Net's decoder). Thus, for each input image-text pair, we end up with a stack of cross-attention features $A \in \mathrm{R}^{T \times L \times D \times S}$, where $T = 30$ diffusion timesteps, $L = 4$ cross-attention layers, $D = 256$ is the feature dimensionality (fixed across the selected layers), and $S$ is the token sequence length (fixed at the maximum length of 77 tokens for CLIP). Note that, for each layer $l = 1, \ldots, L$, the output of the cross-attention operation $A_l$ is

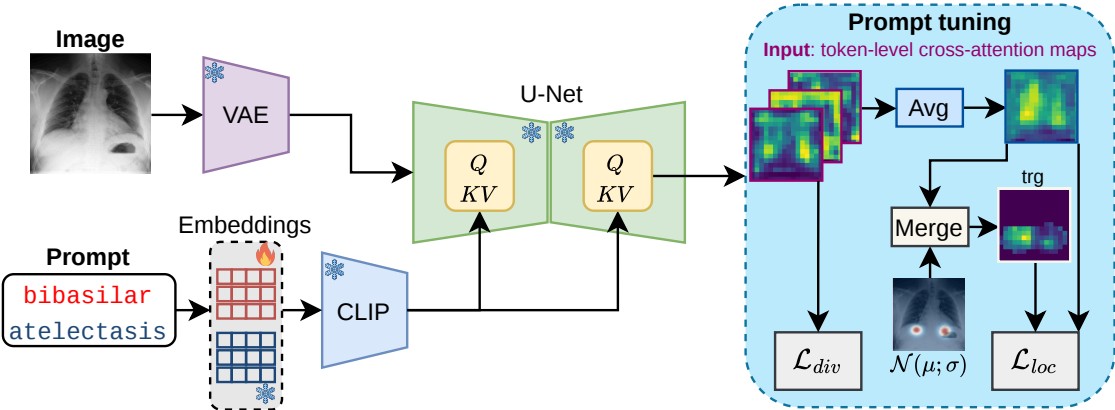

Figure 4: Overview of our proposed prompt tuning method. Given an input image-text pair, we first extract features from intermediate cross-attention layers (highlighted with purple borders). Then, the supervision target $trg$ defined in Equation (3) is dynamically extracted by merging the average (across the token sequence) cross-attention map with the corresponding Gaussian prior $\mathcal{N}(\mu; \sigma)$. The cross-attention features are supervised via the diversity loss $\mathcal{L}_{div}$ and the localization loss $\mathcal{L}_{loc}$ defined in Equations (2) and (4), respectively. Note that only the anatomy token embeddings (in red) are learnable in this setup.

defined according to Equation 1, where queries $Q$ (respectively, keys $K$) are derived from a linear projection of the image (respectively, text) features used as input at layer $l$.

$$A_l\left(Q, K\right) = \text{softmax}\left(\frac{QK^\top}{\sqrt{d}}\right). \tag{1}$$

Our optimization objective consists of two loss terms. The first loss term, called diversity loss $\mathcal{L}_{div}$ is defined by

$$\mathcal{L}_{div} = \mathbb{E}_{t,l}\mathbb{E}_{i,j}\left[\cos^2(A_i^{(l,t)}, A_j^{(l,t)})\right], \quad i \neq j, \, i = 1, \ldots, S, \, t = 1, \ldots, T, \, l = 1, \ldots, L, \tag{2}$$

and is applied to the $\ell_2$-normalized (across the feature dimension $D$) cross-attention features $A$ to minimize the overlap between token-level activations (excluding special tokens).

Then, to enforce the desired spatial behavior to cross-attention activations, we dynamically compute the target supervision signal for each image. More specifically, we first average the cross-attention activations $A$ across the sequence dimension (excluding special tokens) and then reshape those into a spatial feature map $A_{sp} \in \mathbb{R}^{T \times L \times \sqrt{D} \times \sqrt{D}}$. Then, given the Gaussian target $G \in \mathbb{R}^{\sqrt{D} \times \sqrt{D}}$ (reshaped to match the spatial dimensions of $A_{sp}$) which was derived from the data curation process, we define the supervision target $trg$ as

$$trg = \mathbb{1}_{G > \epsilon} \cdot \text{sg}(A_{sp}) + \alpha G, \tag{3}$$

where $\mathbb{1}_{G > \epsilon}$ is a binary mask derived from the Gaussian $G$, $\text{sg}()$ refers to the stop gradient operator (meaning that we detach $A_{sp}$ from the computation graph to calculate $trg$), while the second term acts as a regularizer in the case where the anatomical location specified

in text does not receive any cross-attention activations. In practice, we set $\epsilon = 10^{-5}$ and $\alpha = 0.1$. The second loss term, called localization loss $\mathcal{L}_{loc}$ defined as

$$\mathcal{L}_{loc} = \mathbb{E}[1 - cos(A_{sp}, trg)], \tag{4}$$

aims to align the direction of $A_{sp}$ with that of the target $trg$ after $\ell 2$-normalization across the feature dimension. The final optimization objective is defined as $\mathcal{L} = \mathcal{L}_{div} + \mathcal{L}_{loc}$. An overview of our proposed method is presented in Figure 4.

## 4. Experiments

To validate our approach, we conduct a series of experiments comparing our model's performance against established methods. In this section, we first outline the experimental setup—including datasets, metrics, and baselines—followed by an analysis of the results.

### 4.1. Datasets

We evaluate our proposed system on an established phrase grounding benchmark dataset called MS-CXR (Boecking et al., 2022). Specifically, for consistency across setups, we transform the original prompts in MS-CXR into the standardized "{location} {pathology}" format using RadGraph-XL, and discard (image, text) pairs for which the RadGraph-XL model did not predict any "ANAT-DP" label. We refer to this subset as **MS-CXR-loc** (see Appendix I for an analysis of RadGraph-XL's errors on MS-CXR).

Furthermore, to include an OOD dataset (with respect to the imaging domain) in our analysis, we also evaluate performance on **VinDr-CXR** (Nguyen et al., 2022) test set augmented with synthetic prompts that we derive from ground truth bounding box annotations. We describe how we generate synthetic prompts for VinDr-CXR in Appendix E.

### 4.2. Metrics

Following standard practice (Boecking et al., 2022; Bannur et al., 2023), we evaluate performance on the phrase grounding task based on two metrics:
**Contrast-to-Noise Ratio (CNR)**, which measures the statistical difference of the heatmap distribution within ($A$) and outside ($\bar{A}$) the ground truth bounding box area.

$$CNR = \frac{\mu_A - \mu_{\bar{A}}}{\sqrt{\sigma_A^2 + \sigma_{\bar{A}}^2}}. \tag{5}$$

**Mean Intersection over Union (mIoU)**, which measures the overlap between the ground truth bounding box area and the thresholded heatmap. In our analysis, we select 5 threshold values evenly distributed in [0.1, 0.5] range.

### 4.3. Baseline models and heatmap extraction

We compare our fine-tuned LDM with various baselines that establish the SOTA in medical phrase grounding. Note that, in all cases, we use publicly available model checkpoints (source links provided in Appendix G) pre-trained on MIMIC-CXR (Johnson et al., 2019).

**BioViL** (Boecking et al., 2022), which aligns image and text modalities (via late fusion) using both local and global contrastive losses. The predicted heatmap is defined as the cosine similarity between image and text features projected onto the shared latent space.

**BioViL-T** (Bannur et al., 2023), which extends the BioViL method by introducing a spatio-temporal pre-training task based on cases that contain both a prior and a current study. Note that BioViL-T shares the same heatmap prediction pipeline as BioViL.

**LDM (frozen)** (Pinaya et al., 2022), which refers to the original LDM implementation (trained on PA-view CXRs) with frozen CLIP text embeddings. For heatmap extraction, we use the method introduced in (Vilouras et al., 2024) to extract a heatmap from intermediate cross-attention layers ($\{3, 4, 6, 7\}$ out of 11 layers in total) and diffusion timesteps ($t \in [40, 60]$ out of 100 inference steps), followed by binary Otsu thresholding to suppress background heatmap activations.

**LDM (ours)** denotes the frozen LDM after applying our proposed prompt tuning method. While we follow the same heatmap extraction method as the frozen baseline, we select a different range of cross-attention layers ($\{4, 5, 6, 7\}$) to capture the features learned during prompt tuning, and omit the Otsu thresholding step.

In terms of the evaluation protocol, all heatmaps are first min-max normalized to [0,1]. Since each model outputs a heatmap at a different spatial resolution, we then upsample the predicted heatmaps to the image's shortest side and pad them to the full input resolution prior to evaluation.

### 4.4. Implementation details

For the fine-tuning process, we used the following hyperparameters: batch size $= 1$, learning rate $= 10^{-4}$, range of diffusion timesteps used for extracting cross-attention features $T = [30, 60]$ (after setting the LDM to inference mode with a total of 100 steps), number of cross-attention layers $L = 4$, and $\alpha = 0.1$ (defined in Equation 3). To justify the choice for hyperparameter $\alpha$, we also provide a dedicated ablation study in Appendix H.2. Moreover, we introduce stochasticity in the anatomical Gaussian priors during fine-tuning by randomly varying $\mu$ and $\sigma$ by $\pm 5\%$ of the input image resolution (i.e., by $\pm 25$ pixels for $512 \times 512$ CXRs). Note that optimization takes $\sim 3$ hours on an RTX 3090 GPU with 24GB of RAM.

### 4.5. Results

Phrase grounding results on the **MS-CXR-loc** dataset are shown in Table 1, whereas for **VinDr-CXR** are presented in Table 2. For each metric, we report the per-class mean along with the 95% confidence intervals calculated with bootstrapping (10k resamples). Note that the reported results correspond to a single run. We also provide results for two additional methods in Appendix F, i.e., a large vision-language model called MAIRA-2 (Bannur et al., 2024) that used 50% of MS-CXR data during training, and an oracle method that generates heatmaps based on the ground truth annotations of each evaluation dataset.

#### 4.5.1. DISCUSSION

Based on the results shown in Table 1 and Table 2, respectively, our method yields SOTA phrase grounding performance on both benchmarks, outperforming strong baselines trained with contrastive learning in most pathologies by a significant margin. More specifically,

based on non-overlapping bootstrapped 95% confidence intervals, we observe statistically significant improvements over the strongest baseline for five pathologies (Pneumonia, Pneumothorax, Atelectasis, Edema and Cardiomegaly). These trends suggest that performance gains vary across pathologies and do not follow a consistent pattern based on the size or the spatial extent of the pathology alone. Notably, our method substantially improves on *Pneumothorax* compared to the frozen LDM baseline, highlighting the value of location modifiers for improving downstream performance. Moreover, our LDM even outperforms MAIRA-2 on VinDr-CXR (cf. Appendix F).

Table 1: Phrase grounding results on **MS-CXR-loc** dataset. Best metrics per-class are highlighted with **bold**, second best are underlined. * denotes significant improvement over the best baseline (based on non-overlapping 95% CIs).

| Models / Classes | BioViL | | BioViL-T | | LDM (frozen) | | LDM (ours) | |
|---|---|---|---|---|---|---|---|---|
| | **CNR** | **mIoU (%)** | **CNR** | **mIoU (%)** | **CNR** | **mIoU (%)** | **CNR** | **mIoU (%)** |
| Pneumonia | 1.57 | 27.9 | **1.69** | 29.0 | 1.19 | 21.6 | 1.64 | **36.9*** |
| (N=146) | [1.43, 1.71] | [25.3, 30.4] | [1.57, 1.80] | [26.9, 31.0] | [1.10, 1.29] | [19.9, 23.7] | [1.52, 1.77] | [34.5, 39.2] |
| Pneumothorax | 0.61 | 10.1 | 0.93 | 13.0 | -0.07 | 6.10 | **1.38*** | **15.4** |
| (N=204) | [0.51, 0.70] | [8.76, 11.6] | [0.83, 1.04] | [11.3, 15.0] | [-0.16, 0.03] | [4.93, 7.60] | [1.21, 1.55] | [13.6, 17.4] |
| Consolidation | 1.80 | 29.5 | **1.89** | **30.4** | 1.24 | 23.3 | 1.38 | 27.0 |
| (N=105) | [1.64, 1.97] | [26.8, 32.2] | [1.74, 2.03] | [28.1, 32.6] | [1.11, 1.37] | [20.4, 26.5] | [1.22, 1.55] | [24.6, 29.4] |
| Atelectasis | 0.73 | 10.7 | 1.14 | 13.3 | 1.08 | 21.5 | **1.45** | **32.7*** |
| (N=55) | [0.52, 0.96] | [8.13, 13.7] | [0.96, 1.33] | [10.7, 16.2] | [0.95, 1.20] | [18.7, 24.9] | [1.28, 1.63] | [28.8, 36.5] |
| Edema | 0.74 | 18.0 | 0.74 | 17.6 | 0.84 | **30.5** | **1.16*** | 28.4 |
| (N=41) | [0.57, 0.94] | [14.1, 22.8] | [0.57, 0.89] | [13.9, 22.3] | [0.69, 0.98] | [24.9, 36.4] | [1.00, 1.34] | [23.5, 33.7] |
| Cardiomegaly | 0.73 | 22.1 | 1.04 | 23.2 | **1.16** | 40.8 | 1.13 | **43.8*** |
| (N=333) | [0.66, 0.79] | [20.1, 24.0] | [0.99, 1.10] | [21.6, 24.9] | [1.10, 1.23] | [39.6, 42.1] | [1.10, 1.16] | [42.8, 44.7] |
| Lung Opacity | 1.58 | 19.0 | **1.77** | 21.4 | 1.11 | 15.6 | 1.58 | **24.0** |
| (N=78) | [1.37, 1.80] | [16.3, 22.2] | [1.58, 1.97] | [18.8, 24.1] | [0.98, 1.25] | [13.1, 18.4] | [1.40, 1.80] | [21.1, 27.3] |
| Pleural Effusion | 1.60 | 24.5 | **1.75** | 23.3 | 0.88 | 16.5 | 1.22 | **28.1** |
| (N=81) | [1.43, 1.76] | [22.3, 26.6] | [1.59, 1.91] | [21.0, 25.4] | [0.74, 1.01] | [14.4, 18.9] | [1.09, 1.35] | [25.3, 30.9] |
| Average | 1.17 | 20.2 | **1.37** | 21.4 | 0.93 | 22.0 | **1.37** | **29.5** |

## 4.6. Additional results

To complement our main results, we include additional experiments in the Appendix. A brief description of each experiment and key takeaways are provided below.

First, in Appendix H.1 we evaluate whether the standardized "{location} {pathology}" prompt format—which omits other modifiers such as the severity level—impacts any of the baselines used in our study. Interestingly, compared to free-form text (i.e., sentences extracted from the radiology report that describe the finding), performance with the standardized format remained stable on average, with only minor fluctuations per class. This finding suggests that baselines largely ignore other types of modifiers (e.g., severity) present in free-form text.

Second, we conducted a pilot qualitative study involving two radiologists to assess whether improvements in quantitative metrics align with their judgments. The setup and findings of this study are presented in Appendix J. We observed that both CNR and mIoU

Table 2: Phrase grounding results on **VinDr-CXR** test set. Best metrics per-class are highlighted with **bold**, second best are underlined. * denotes significant improvement over the best baseline (based on non-overlapping 95% CIs).

| Models / Classes | BioViL CNR | BioViL mIoU (%) | BioViL-T CNR | BioViL-T mIoU (%) | LDM (frozen) CNR | LDM (frozen) mIoU (%) | LDM (ours) CNR | LDM (ours) mIoU (%) |
|---|---|---|---|---|---|---|---|---|
| Pneumonia | **1.64** | **26.1** | 1.30 | 21.5 | 1.18 | 21.1 | 1.44 | 24.8 |
| (N=31) | [1.38, 1.93] | [21.5, 30.6] | [0.96, 1.64] | [17.0, 25.7] | [0.93, 1.42] | [16.3, 27.7] | [1.15, 1.76] | [21.2, 28.8] |
| Pneumothorax | 1.18 | 9.46 | 1.44 | 10.5 | 0.28 | 5.52 | **1.92** | **17.1** |
| (N=8) | [0.85, 1.35] | [6.02, 13.3] | [1.06, 1.72] | [5.74, 15.4] | [-0.19, 0.92] | [2.14, 10.4] | [1.42, 2.71] | [11.5, 23.4] |
| Consolidation | **2.32** | 20.0 | **2.32** | **21.0** | 1.19 | 9.41 | 2.30 | 20.2 |
| (N=51) | [2.06, 2.57] | [16.9, 23.5] | [2.13, 2.50] | [18.2, 24.0] | [0.94, 1.42] | [7.20, 12.6] | [2.07, 2.55] | [17.5, 23.3] |
| Atelectasis | 1.15 | 3.20 | 1.65 | 4.88 | 1.12 | 4.35 | **2.35*** | **9.71*** |
| (N=60) | [0.92, 1.38] | [2.24, 4.59] | [1.41, 1.87] | [3.63, 6.56] | [0.91, 1.32] | [2.95, 7.05] | [2.13, 2.57] | [7.61, 12.4] |
| Cardiomegaly | 0.88 | 18.7 | **1.50** | 23.3 | 1.25 | 27.7 | 1.47 | **42.8*** |
| (N=189) | [0.81, 0.96] | [16.8, 20.7] | [1.43, 1.58] | [21.5, 25.3] | [1.14, 1.38] | [26.1, 29.4] | [1.42, 1.52] | [41.4, 44.1] |
| Lung Opacity | 1.50 | 8.12 | 1.39 | 6.94 | 0.93 | 4.69 | **2.05** | **11.1** |
| (N=48) | [1.19, 1.83] | [6.06, 11.0] | [1.14, 1.65] | [5.25, 8.87] | [0.64, 1.19] | [3.70, 5.83] | [1.79, 2.30] | [9.16, 13.4] |
| Pleural Effusion | 1.68 | 15.9 | **1.75** | 16.1 | 0.84 | 9.61 | 1.10 | **17.8** |
| (N=56) | [1.46, 1.91] | [13.0, 19.0] | [1.53, 1.99] | [13.3, 19.2] | [0.62, 1.05] | [7.48, 12.6] | [0.93, 1.30] | [14.2, 21.5] |
| Average | 1.48 | 14.5 | 1.62 | 14.9 | 0.97 | 11.8 | **1.80** | **20.5** |

correlate significantly with radiologists' ratings, and instance-level improvements in these metrics reflect perceptually meaningful differences.

Third, in Appendix K we demonstrate the downstream utility of heatmaps through the task of grounded pneumothorax classification. Our analysis shows that more accurate heatmap localisation corresponds to higher classification performance, highlighting the practical relevance of our approach.

## 5. Conclusion

In this work, we propose an efficient fine-tuning method to improve multi-modal alignment within a pre-trained text-to-image Latent Diffusion Model. We present a novel data curation pipeline to extract a weak supervision signal based on references of anatomical locations in unstructured radiology reports. In turn, we use this coarse signal to steer the cross-attention activations of the pre-trained model towards the correct anatomical area by fine-tuning the text embeddings with a small subset ($\sim 6,500$ samples) of the original training set. Last, we show that our method improves image-text alignment by evaluating on the phrase grounding task, where our model achieves SOTA performance on an established benchmark dataset (MS-CXR), as well as on OOD data (VinDr-CXR). These findings are further supported by qualitative analyses and an additional grounded disease classification experiment (for a discussion of limitations and ethical considerations, please refer to Appendix A and Appendix B, respectively). We also note that, since the RadGraph-XL model was trained on four anatomy-modality pairs (including CT and MRI), our approach can be extended to these modalities provided that domain-specific latent diffusion models are available. Beyond the present study, our approach could be applied to generate pseudo-masks

for downstream model training, or used as an auxiliary task in general vision-language pre-training.

## Acknowledgments

This work was supported by the University of Edinburgh by PhD studentships to Konstantinos Vilouras, Ilias Stogiannidis, and Junyu Yan. Sotirios A. Tsaftaris also acknowledges support by a Canon Medical / Royal Academy of Engineering Research Chair under Grant RCSRF1819\8\25.

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

## Appendix A. Limitations

We identified the following limitations regarding our work:

First, our proposed method assumes that radiology reports provide explicit references to both the pathology and its corresponding anatomical location, yet these details are not always consistently reported in practice. Additionally, our method's effectiveness depends on RadGraph-XL's ability to accurately extract anatomy-related tokens from free-form text, which may fail in certain cases (see Appendix I for an error analysis on MS-CXR).

Second, in our analysis, we discard useful clinical information specified in text that provides a more fine-grained description of the underlying pathology, e.g., descriptors of the pathology's distribution across the lungs such as *diffuse* or *focal*, and severity modifiers such as *mild* or *acute* (which could be used in future work to adaptively control the size and intensity of heatmaps). Moreover, the list of anatomical locations used in this study is by no means exhaustive. Future research could benefit from addressing these limitations.

Last, the LDM used in this work follows the same text processing pipeline (i.e., the CLIP tokenizer and text encoder) as Stable Diffusion 2.1. As a result, since CLIP has

not encountered radiology reports during training, specialized terms (e.g., words indicating pathologies) are split into multiple sub-tokens.

## Appendix B. Ethics statement

The two datasets used in this study, i.e., MIMIC-CXR (Johnson et al., 2019) and VinDr-CXR (Wu et al., 2021), are publicly available with credentialed access (according to PhysioNet Credentialed Health Data License 1.5.0) through PhysioNet[2] and have been de-identified prior to their release. Furthermore, our proposed system has not been extensively tested for its diagnostic accuracy in diverse settings, or for the presence of biases, which could have significant implications and pose potential risks, e.g., for fairness across different demographic groups. Therefore, our model is by no means ready for deployment in real-world clinical practice.

## Appendix C. Extended Related Works

Prior work in computer vision, such as Zhu et al., 2017, has incorporated spatial regularization constraints for classification by applying learnable convolutions over label-specific attention maps. Although this approach captures spatial relations among labels, the modeling remains implicit and is not guided by external spatial or textual cues. On the contrary, related multi-modal methods (Li et al., 2023b) leverage image-text data along with a set of expert-annotated masks for semi-supervised segmentation. Our method, however, adopts a weakly supervised approach based on report-level supervision for phrase grounding, thereby eliminating the need for fine-grained, pixel-level annotations. More recently, context optimization (Zhou et al., 2022) improved the few-shot classification performance of frozen CLIP-style models by prepending a fixed set of learnable context tokens to text prompts. However, these tokens do not explicitly consider linguistic semantics that could improve image-text alignment. Another line of work explores prompt tuning for diffusion-based image editing; for instance, Dong et al., 2023 optimize global (image-level) conditional embeddings to balance the trade-off between editability and fidelity. While effective for global control, these embeddings lack spatial or semantic grounding. In contrast, our approach fine-tunes anatomy-specific token embeddings associated with explicit location entities mentioned in free-text reports. This allows the model to steer cross-attention activations toward semantically relevant image regions, providing spatially grounded localization rather than relying on global prompt optimization or attention regularization techniques.

## Appendix D. Details on data curation

Here, we provide a detailed explanation of our proposed data curation process. First, we share details about our rule-based post-processing step aiming to derive a fixed set of location tokens from RadGraph-XL's predictions. Then, we explain how we map location tokens to a weak supervision signal (2D Gaussian) based on a small set of annotations from the Chest Imagenome dataset (Wu et al., 2021).

---

2. https://physionet.org/

From the original dataset used to train the LDM ($\sim$ 70k samples), we first create a subset consisting of the 8 pathologies of interest (Pneumonia, Pneumothorax, Pleural Effusion, Lung Opacity, Atelectasis, Cardiomegaly, Consolidation, Edema), where each sample is assigned to exactly one pathology. Then, we search for sentences in the accompanying reports that refer to the ground truth pathology while accounting for the plural form of each term, and also for synonyms (e.g., radiologists might use the terms *opacity* or *consolidation* interchangeably with the term *pneumonia*). Note that we also skip sentences that refer to the ground truth label while containing words indicating negation (e.g., *no* or *without*) or resolution (e.g., *resolved*). Moreover, we develop a rule applied on both the sentence and the report level to merge bilaterally symmetrical regions into a single location term (e.g., {"left lower", "right lower"} → "bilateral lower"). As a result, we end up with the following 27 locations in total:

- {"left", "right", "bilateral"} × {"", "apical", "upper", "middle", "lower", "costophrenic", "pleural", "base"}, where "" denotes the empty string and × denotes the cartesian product. Note also that the phrase "bilateral base" is mapped to the word "bibasilar".

- "lingular", to indicate the lower area of the left lung's middle lobe.

- "cardiomegaly", to indicate the area of the increased heart.

- "pulmonary", which is commonly used for the label Edema and refers to the area of both lungs.

Furthermore, to derive the coarse supervision signals, we rely on the gold standard subset of Chest Imagenome dataset that contains 1,000 images with bounding box annotations per anatomical area. Note that the overlap between Chest Imagenome and our fine-tuning set is small (28 images out of 6,480 used for fine-tuning in total), thus the supervision signals are considered weak. With respect to our predefined set of anatomical locations, since Chest Imagenome lacks annotations for some terms, we also perform the following mappings {"base" → "lower lung zone", "pleural" → "lower lung zone", "lingular" → "left mid lung zone"} to ensure that each of the 27 locations mentioned above is assigned to a Gaussian. Moreover, since the LDM expects a fixed size image with dimensions $512 \times 512$ as input, we transform the ground truth bounding boxes to the same resolution. Then, for each bounding box, we calculate the mean $\mu$ and standard deviation $\sigma$ per spatial dimension using Equation (6):

$$
\begin{aligned}
\mu_x &= \frac{x_{min} + x_{max}}{2}, & \sigma_x &= (x_{max} - x_{min})/6 \\
\mu_y &= \frac{y_{min} + y_{max}}{2}, & \sigma_y &= (y_{max} - y_{min})/6,
\end{aligned}
\tag{6}
$$

where the formula for standard deviation is set according to the $\pm 3\sigma$ rule for Gaussians, meaning that $\sim$ 99.7% of activations is within the specified min-max range. Last, we average the bounding box statistics per anatomical location to derive the fixed set of parameters for each 2D Gaussian. Figure 3 illustrates the final set of 2D Gaussians for each of the anatomical areas provided by Chest Imagenome.

## Appendix E.  Generating synthetic prompts for VinDr-CXR

All models considered in this study were trained on subsets of the large MIMIC-CXR database (Johnson et al., 2019). Moreover, MS-CXR (Boecking et al., 2022), which is the only publicly available phrase grounding benchmark, is part of the same database. Therefore, in an attempt to evaluate models on an OOD dataset (i.e., with CXRs from a different hospital), we propose a simple heuristic method to augment VinDr-CXR (Wu et al., 2021) samples with synthetic prompts (following the ``{location} {pathology}'' format) derived from ground truth bounding box coordinates. More specifically, after extracting the statistics $(\mu_x, \mu_y, \sigma_x, \sigma_y)$ of each bounding box in VinDr-CXR using Equation (6), we identify the Gaussian from our LUT (also shown in Figure 3) that is closest to the given bounding box based on the squared 2-Wasserstein distance metric presented in Equation (7)

$$\operatorname*{argmin}_{i} \left[ (\mu_x - \mu_x^i)^2 + \ (\mu_y - \mu_y^i)^2 + \ (\sigma_x - \sigma_x^i)^2 + \ (\sigma_y - \sigma_y^i)^2 \right], \tag{7}$$

where $i = 1, ..., N$ refers to the total number of Gaussians available in our LUT. As a result, the synthetic prompt is formed using the retrieved location from our LUT and the image's ground truth pathology label. Note also that, in the case where an image has more than 1 bounding box, we bind the retrieved locations using the ⟨and⟩ token.

## Appendix F.  Additional phrase grounding results

For completeness, we also report phrase grounding results for the following two baseline methods:

**MAIRA-2** (Bannur et al., 2024), a recently released multi-modal large language model. Although tailored to the task of radiology report generation, the authors also provide guidelines on how to use the model to perform phrase grounding. Note, however, that MAIRA-2 was trained on 50% of MS-CXR data, thus a direct comparison with other baselines would not be fair. It is also worth mentioning that MAIRA-2 outputs bounding box coordinates (instead of a heatmap), therefore we only use the mIoU metric to report results in Table 3.

We observe that MAIRA-2 clearly underperforms in the OOD scenario, since most baselines (shown in Table 2) achieve a higher average mIoU across classes. Moreover, performance on *Pneumothorax* and *Cardiomegaly* is consistently high in all setups. It is also worth mentioning that, during evaluation on VinDr-CXR, MAIRA-2 did not predict a bounding box for 129 (out of 443 in total) input image-text pairs. More specifically, the number of times MAIRA-2 did not provide a prediction per class is: {*Atelectasis*: 9, *Cardiomegaly*: 26, *Consolidation*: 34, *Lung Opacity*: 24, *Pleural effusion*: 18, *Pneumonia*: 18, *Pneumothorax*: 0}.

**Oracle Gaussian**, where we use the ground truth bounding box coordinates to generate an optimal heatmap following a Gaussian distribution. The parameters of the Gaussian for the x-coordinate (respectively for the y-coordinate) are computed as $\mu_x = \ (x_{max} + x_{min})/2$, $\sigma_x = \ (x_{max} - x_{min})/3$. This method can be interpreted as the empirical upper bound on phrase grounding performance per evaluation dataset. Results are shown in Table 4.

Table 3: Phrase grounding results for the **MAIRA-2** model (Bannur et al., 2024) based on the mIoU metric (in %). Note that the test set of VinDr-CXR does not contain any samples with label Edema.

| Classes | MS-CXR-loc | VinDr-CXR |
|---|---|---|
| Pneumonia | 45.6 | 5.57 |
| | [42.3, 48.8] | [3.10, 9.01] |
| Pneumothorax | 45.1 | 14.5 |
| | [41.7, 48.0] | [5,61, 23.3] |
| Consolidation | 35.0 | 6.71 |
| | [31.0, 38.8] | [4.20, 10.2] |
| Atelectasis | 37.0 | 5.69 |
| | [32.3, 42.9] | [3.76, 9.66] |
| Edema | 27.8 | N/A |
| | [22.5, 32.1] | |
| Cardiomegaly | 78.1 | 52.5 |
| | [77.1, 79.1] | [50.1, 54.9] |
| Lung Opacity | 32.2 | 5.62 |
| | [32.2, 41.4] | [3.70, 8.48] |
| Pleural Effusion | 42.7 | 8.16 |
| | [38.8, 47.7] | [4.10, 14.5] |
| Average | 43.5 | 14.1 |

Table 4: Phrase grounding results for the **Oracle Gaussian** method. We only report average results since variability across classes is low for this approach.

| Datasets | Metrics | |
|---|---|---|
| | Avg CNR | Avg mIoU (%) |
| MS-CXR-loc | 2.11 | 67.6 |
| VinDr-CXR | 2.23 | 67.5 |

## Appendix G. Model sources and licenses

Here, we list the sources and licenses for all publicly available models used in our study:

- RadGraph-XL (Delbrouck et al., 2024): Available through `https://pypi.org/project/radgraph/` (MIT License)

- BioViL (Boecking et al., 2022) and BioViL-T (Bannur et al., 2023): Checkpoints for these two models are available in `https://huggingface.co/microsoft/BiomedVLP-BioViL-T` (MIT License)

- MAIRA-2 (Bannur et al., 2024): Available in `https://huggingface.co/microsoft/maira-2` (Microsoft Research License Agreement)

Table 5: Ablation study for the hyperparameter $\alpha$ introduced in Equation (3). For this experiment, we only provide results on the **MS-CXR-loc** dataset.

| Setup | Metrics | |
|---|---|---|
| | **Avg CNR** | **Avg mIoU (%)** |
| $\alpha = 0$ | 1.30 | 28.5 |
| $\alpha = 1$ | 1.18 | 24.6 |
| $\alpha = 0.1$ (ours) | **1.37** | **29.5** |

- LDM (Pinaya et al., 2022): Pre-trained weights are available in `https://github.com/Project-MONAI/GenerativeModels/tree/main/model-zoo/models/cxr_image_synthesis_latent_diffusion_model` (Apache 2.0 License)

## Appendix H. Additional experiments

### H.1. Impact of prompt format on phrase grounding performance

Throughout this work, we report phrase grounding results based on the structured "`{location}` `{pathology}`" prompt format. However, to facilitate comparison with prior approaches that use free-text sentences as prompts, we conduct an additional experiment to measure how the input prompt format affects the pre-trained models' downstream performance. The results are depicted in Figure 5. Overall, we observe that the CNR metric remains stable on average. On the contrary, although the average mIoU is not significantly affected (in the worst-case scenario, switching to the standardized prompt format leads to a decrease in BioViL's performance by 1.2%), it is clear that results across classes fluctuate. Most notably, for the BioViL(-T) baselines, mIoU for label *Atelectasis* decreases by more than 10%, whereas for *Lung Opacity* and *Pleural Effusion* it increases by at least 3.5%. However, in the case of the frozen LDM, we notice that the standardized prompt format mostly improves phrase grounding results by a small amount.

### H.2. Ablation study

To justify our choice of $\alpha = 0.1$ for the hyperparameter $\alpha$ defined in Equation (3), we provide a separate ablation study in Table 5. Furthermore, through an ablation study on the $\mathcal{L}_{div}$ loss term defined in Equation (2), we observed that performance on the MS-CXR-loc dataset slightly increases on average when we remove this term (0.01 in terms of CNR and 0.4% in terms of mIoU). Upon further inspection, we observe that this improvement in performance is mostly due to the *Pneumothorax* class, while metrics for most of the other classes dropped slightly. We speculate that this behavior can be attributed to the following: First, pneumothorax (which occurs when lung punctures, i.e., dark air escapes into pleural space, whilst the white lung collapses inwards) is fundamentally different compared to other pathologies (which manifest as white regions either in or around the lung area). Second, the fact that some pathologies tend to appear in specific anatomic areas (e.g., pneumothorax usually appears at the apex of the lungs) could also possibly contribute towards this discrepancy in results. We leave further investigation for future work. In addition, we evaluated the sensitivity of the loss $\mathcal{L} = \lambda_{div}\mathcal{L}_{div} + \lambda_{loc}\mathcal{L}_{loc}$ by ablating the

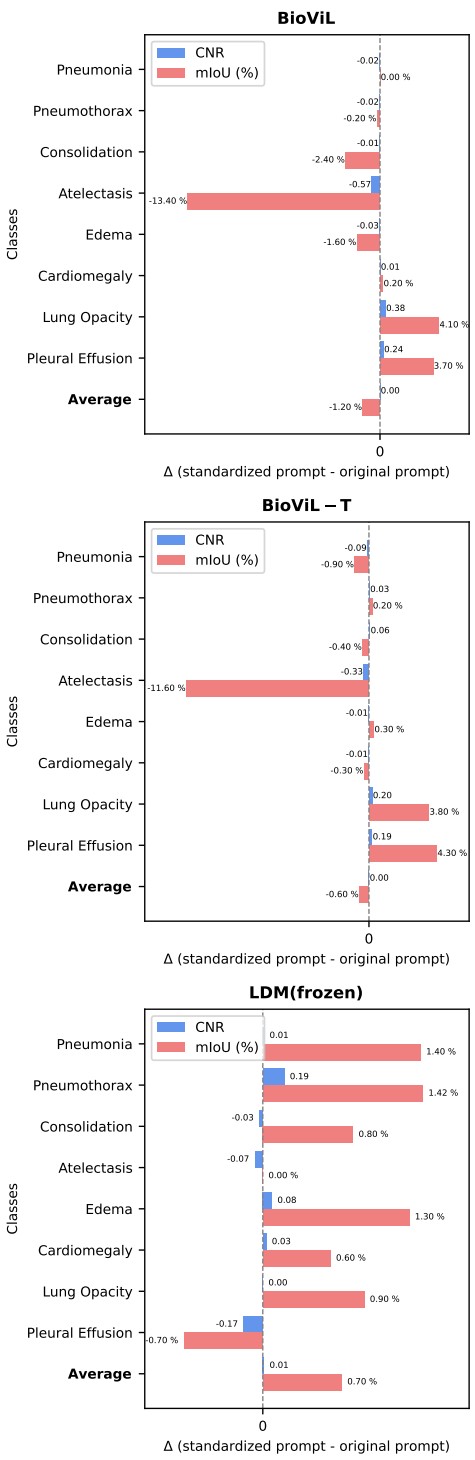

Figure 5: Impact of prompt format on phrase grounding performance. For this analysis, we use the MS-CXR dataset. We report the difference in terms of per-class mean metrics between the standardized "{location} {pathology}" and the original (free-text) prompt format.

weighting coefficients $\lambda_{div}$ and $\lambda_{loc}$ on the MS-CXR-loc dataset. For this experiment, we used four different setups, i.e., ($\lambda_{div} = 2$, $\lambda_{loc} = 1$), ($\lambda_{div} = 10$, $\lambda_{loc} = 1$), ($\lambda_{div} = 1$, $\lambda_{loc} = 2$), and ($\lambda_{div} = 1$, $\lambda_{loc} = 10$). Compared to our best performing setup ($\lambda_{div} = \lambda_{loc} = 1$, results shown in Table 1), the least optimal configuration ($\lambda_{div} = 10, \lambda_{loc} = 1$) resulted in a marginal performance drop of 0.06 in terms of CNR and $\sim 2\%$ in terms of mIoU. Overall, setting $\lambda_{loc} \geq \lambda_{div} > 0$ proved to be optimal in this task.

## Appendix I. RadGraph-XL error analysis

In this section, we present an error analysis on the MS-CXR dataset to identify the failure modes of RadGraph-XL in extracting anatomical location tokens. More specifically, since we did not observe any false positives (i.e., cases where a token was incorrectly labeled as "ANAT-DP"), our analysis is focused on false negatives. It is also worth mentioning that the presence of false negatives does not compromise either the fine-tuning or the evaluation protocol, since those samples are automatically removed from the datasets (due to the absence of a predicted "ANAT-DP" token). Moreover, we report that the following cases were excluded from our analysis: (a) prompts that do not contain anatomy-related information, e.g., "*pneumothorax*", (b) sentences where lung laterality (right or left) was not explicitly specified, e.g., "*basilar atelectasis*", (c) predicted anatomical areas outside the scope of our pre-defined dictionary of locations (see Appendix D), e.g., "*increased parenchymal opacity in the retrocardiac region*". As a result, 117 out of the 1,160 image-text pairs in MS-CXR were excluded from our evaluation. From those 117 cases, 53 prompts contained anatomy-related information yet they were not assigned an "ANAT-DP" label by RadGraph-XL. These cases are flagged as false negative predictions. The number of false negatives per pathology is the following: {*Atelectasis*: 3, *Edema*: 2, *Cardiomegaly*: 0, *Consolidation*: 3, *Lung Opacity*: 4, *Pleural effusion*: 14, *Pneumonia*: 13, *Pneumothorax*: 14}. Moreover, we identified the following error patterns:

- RadGraph-XL systematically fails to predict the anatomical location mentioned in the prompts "*left pneumothorax*" and "*right pneumothorax*".

- Also, RadGraph-XL's ability to predict locations mentioned at the end of the prompt is surprisingly low, indicating a potential positional bias. Examples of such prompts are the following: "*consolidation at both lung bases*", "*patchy perihilar opacities bilaterally*", "*small pleural effusion is now present on the right*", "*trace pneumothorax on the left*", "*small pleural effusions are present bilaterally*".

## Appendix J. Qualitative study

To complement our quantitative results presented in Section 4.5, we conducted an exploratory human evaluation study in which two radiologists were asked to assess the quality of heatmaps produced by our method on a 5-point Likert scale. The radiologists have four years (R1, FRCR-qualified) and three years (R2, radiology registrar) of experience, respectively. The purpose of this evaluation was to determine whether our method yields visual explanations that align with radiologists' interpretation of the underlying pathology. To

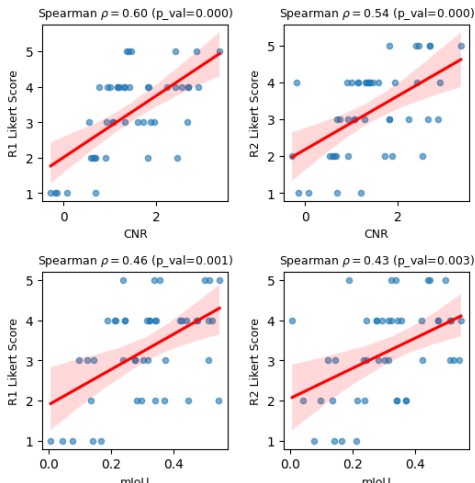

Figure 6: Results of the pilot qualitative study. Note that, on this data subset, our LDM achieved mean CNR=1.44 and mean mIoU=31.2%. The mean score from radiologist 1 (R1) is $3.27 \pm 1.20$, whereas for radiologist 2 (R2) is $3.22 \pm 1.17$.

this end, we randomly selected 50 image-text pairs for each of the 8 pathologies in MS-CXR, ensuring that this subset represents a diverse set of locations per pathology. Note, also, that the ground truth bounding box annotations from MS-CXR were not incorporated into the study interface. For each image-text pair, the radiologists scored the corresponding heatmap based on the following scoring criteria: 1 (very poor) — mostly irrelevant regions highlighted, little or no overlap with relevant areas; 2 (poor) — some correct activation, but primarily irrelevant; 3 (moderate) — partial alignment with relevant regions, but substantial activation on irrelevant areas; 4 (good) — mostly correct, with minor leakage or slight under-representation; 5 (excellent) — predominantly relevant regions highlighted, with minimal to no irrelevant activations. The study was performed on the 3DSlicer[3] platform using the MONAI label[4] extension. As presented in Figure 6, both quantitative metrics correlate significantly with radiologists' ratings (CNR: $\rho = 0.6, 0.54$; mIoU: $\rho = 0.46, 0.43$ for radiologist R1 and R2 respectively, $p << 0.05$), suggesting that more accurate localisation leads to higher perceived heatmap quality.

Furthermore, we examined whether instance-wise improvements in quantitative metrics are consistent with radiologists' assessments. To this end, we collected radiologist scores (following the protocol described above) using the best performing baseline BioViL-T (Bannur et al., 2023) and calculated delta scores for each case, i.e., $\Delta CNR = CNR_{LDM} - CNR_{BioViL-T}$, $\Delta mIoU = mIoU_{LDM} - mIoU_{BioViL-T}$, and $\Delta human = score_{LDM} - score_{BioViL-T}$. We observed a significant positive correlation between $\Delta CNR$ and $\Delta human$ ($\rho = 0.69, p << 0.05$), and also between $\Delta mIoU$ and $\Delta human$ ($\rho = 0.67, p << 0.05$), confirming that quantitative metrics reflect meaningful differences.

---

3. https://www.slicer.org/

4. https://monai.io/label.html

## Appendix K. Grounded classification

To further demonstrate the practical value of improved pathology localisation, we explore grounded classification as a downstream application. Our goal here is to assess the relationship between grounding quality and classification performance. We hypothesize that heatmaps with high localisation accuracy will lead to more accurate grounded classification predictions. Specifically, we focus on the binary task of pneumothorax (PTX) classification using samples from MS-CXR-loc, while for the "healthy" class we construct a balanced subset by randomly sampling data from the MIMIC-CXR test set labeled as "No Finding". Moreover, following the zero-shot diffusion classifier method (Li et al., 2023a), we propose a heatmap-based reweighting of the class-wise prediction error as follows:

$$error_{c_i} = \mathbb{E}_{t,\epsilon} \left[ \frac{h_{c_i} \odot ||\epsilon - \epsilon_\theta(x_t, c_i)||}{\sum h_{c_i}} \right], \tag{8}$$

where $h_{c_i}$ denotes the heatmap for class $c_i$. Note that we use the pre-trained frozen LDM as a classifier here, and the prompts "`No acute cardiopulmonary process`" and "`pneumothorax`" to represent the absence and presence of pneumothorax, respectively. In terms of the heatmap extraction process, we evaluate the following three strategies: (i) frozen LDM using the prompt "`pneumothorax`" in all cases, (ii) frozen LDM with our proposed "`{location} {pathology}`" format, and (iii) our fine-tuned LDM with the same prompt format. Furthermore, since grounded classification requires generating a heatmap for each class to compute weighted prediction errors per test image, we do the following:

- For PTX cases, we generate class-specific heatmaps using the ground truth prompt from MS-CXR (following the standardized format) and the prompt "`No acute cardiopulmonary process`", respectively.

- For each healthy test image, we run multiple independent inference runs. Each run uses a heatmap generated from a different location-specific prompt (i.e., *left apical*, *right apical*, *left*, and *right* PTX locations) along with the heatmap from the base healthy prompt. Each of these location-specific scores is then added to the overall test set pool used to calculate AUROC and the differences in AUROC between low- and high-mIoU bins per heatmap extraction method.

To test our hypothesis that improved heatmap localisation translates into improved classification performance, for each of the three heatmap extraction methods, we split samples into equal-sized low- and high-mIoU bins and measure the difference in AUROC across bins. As shown in Figure 7, the fine-tuned LDM yields a statistically significant result ($\Delta_{\text{AUROC}} = 0.118$, $p < 0.05$) that validates our hypothesis. A detailed breakdown of these classification results is presented in Table 6.

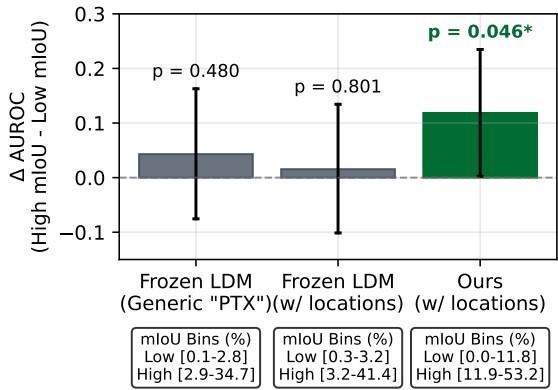

Figure 7: Grounded pneumothorax (PTX) classification results. For each heatmap extraction method, we report the difference in AUROC between high and low mIoU bins (dynamically adjusted for each method to ensure equal-sized bins). Error bars represent 95% CIs. Our proposed method (in green) achieves a statistically significant ($p < 0.05$) positive correlation between localisation quality and classification performance.

Table 6: Detailed grounding classification results. 95% CIs were estimated with bootstrapping (10k resamples). Note that the baseline zero-shot diffusion classifier method (Li et al., 2023a) (i.e., without any heatmaps involved) achieves a mean AUROC of 0.52 in this task.

| Setup | mIoU bin range (%) | AUROC (95% CI) |
|---|---|---|
| Frozen LDM, fixed prompt | Low [0.1, 2.8] | 0.496 (0.414, 0.579) |
| | High [2.9, 34.7] | 0.539 (0.456, 0.622) |
| Frozen LDM, standardized prompt | Low [0.3, 3.2] | 0.427 (0.343, 0.509) |
| | High [3.2, 41.4] | 0.443 (0.361, 0.525) |
| Fine-Tuned LDM (Ours) | Low [0.0, 11.8] | 0.474 (0.391, 0.560) |
| | High [11.9, 53.2] | 0.594 (0.510, 0.674) |

## Appendix L. Additional visualizations

Figures 8 and 9 show qualitative examples of the cross-attention maps corresponding to selected tokens of the input sequence, as well as the average across all tokens, for both the frozen and the fine-tuned LDM. We also show the resulting activations given the original (free-text) and the standardized prompt, respectively. It is clear that the frozen LDM's cross-attentions are not precisely localized, resulting in widely activated regions over the entire image. On the contrary, the fine-tuned LDM yields more fine-grained and spatially accurate cross-attention activations.

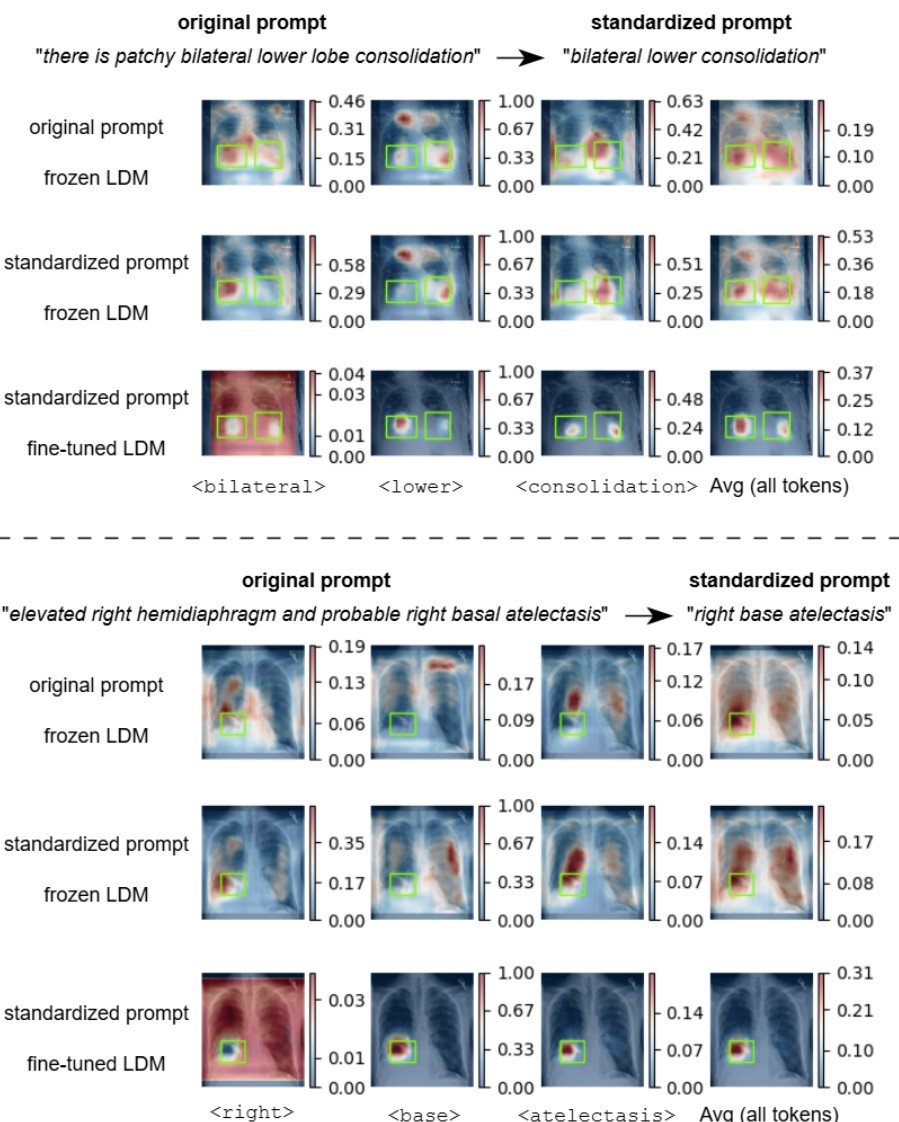

Figure 8: Un-normalized cross-attention visualizations for randomly selected samples from the MS-CXR-loc dataset. For each of the two examples depicted in the figure, we provide the original (free-text) prompt, the standardized prompt, and also mention the version of the LDM used to extract the cross-attentions, i.e., either the frozen or the fine-tuned model. The ground truth bounding boxes of each example are overlayed on top of each heatmap.

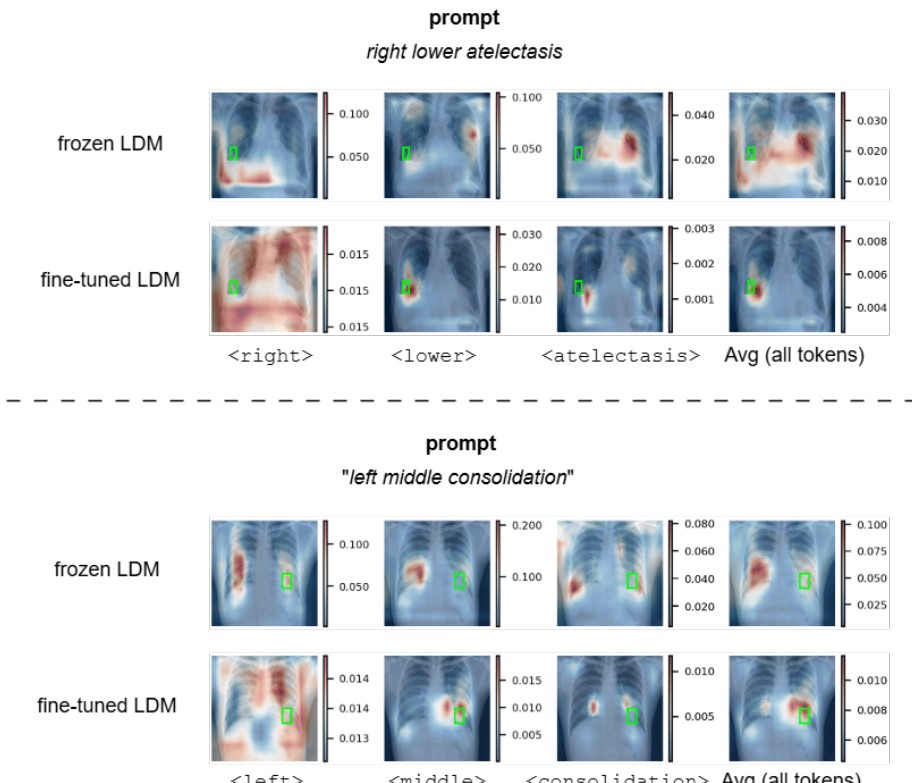

Figure 9: Un-normalized cross-attention visualizations for randomly selected samples from the VinDr-CXR dataset. For each of the two examples depicted in the figure, we provide the synthetic prompt (in the standardized format) and the version of the LDM used to extract the cross-attentions, i.e., either the frozen or the fine-tuned model. The ground truth bounding box of each example is overlaid on top of each heatmap.

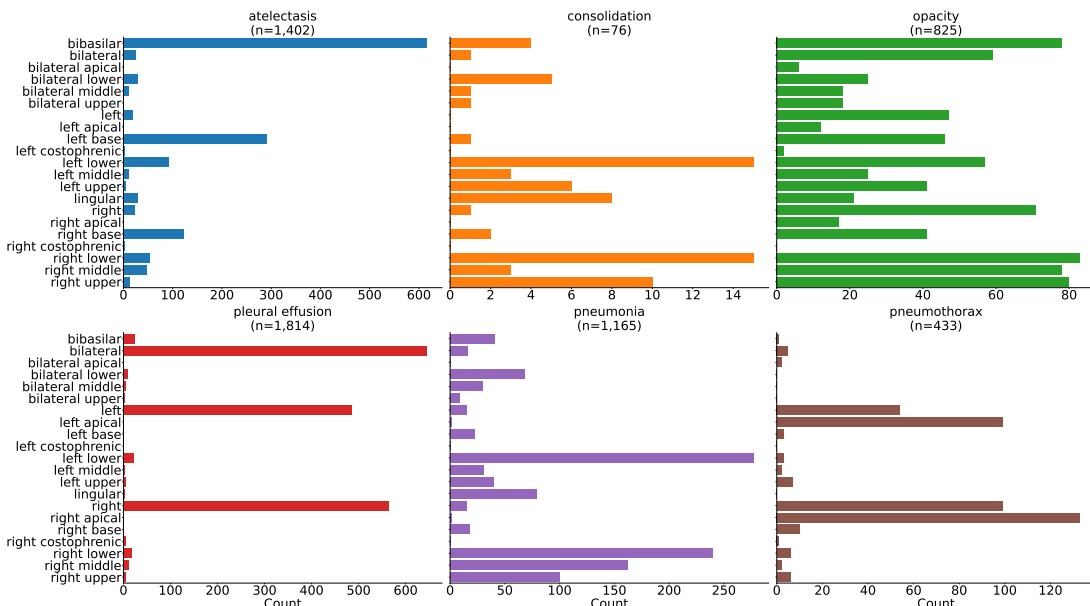

Figure 10: Statistics of the fine-tuning set. For each pathology label, we present the number of samples per anatomical location. Note that we exclude the pathologies *Edema* (N=382 samples) and *Cardiomegaly* (N=1,201 samples) from the figure since those are assigned to exactly one location.

