# OpenReview forum: "Anatomy-Grounded Weakly Supervised Prompt Tuning for Chest X-ray Latent Diffusion Models"
_MIDL.io/2026/Conference — MIDL 2026 Poster_

### Official Review · Reviewer_xmN6 · 2025-12-20

**Confidence:** 4
**Preliminary Rating:** 4
**Final Rating:** 5

**Summary:**

The paper proposes an anatomy-grounded fine-tuning strategy for latent diffusion models (LDMs) in chest X-ray generation that leverages weakly supervised signals derived from radiology reports to improve image–text alignment. Specifically, the method employs clinical entity recognition and a tailored optimization objective to update anatomy-related token embeddings, while keeping the remaining model components frozen, resulting in a parameter-efficient adaptation scheme. By steering cross-attention activations toward anatomically relevant regions associated with pathological findings, the approach aims to enhance spatial grounding without requiring pixel-level supervision. The effectiveness of the method is demonstrated on two benchmark datasets through quantitative evaluations, qualitative analyses, and a downstream experiment on grounded disease classification, indicating improved anatomical localization and clinical relevance.

**Strengths:**

1.	The experimental evaluation is extensive and well structured, covering quantitative results on established benchmark datasets, qualitative visual analyses, and downstream task experiments for grounded disease classification. In addition, the ablation studies provided in the supplementary material offer deeper insight into the contribution of individual components of the proposed method and highlight the overall rigor and scope of the authors’ work.
2.	The reported results show consistent improvements across multiple aspects of phrase grounding and suggest robustness to domain shifts, as indicated by the performance on an out-of-distribution dataset. This robustness is particularly valuable for medical imaging applications, where variations in acquisition protocols and patient populations are common.
3.	The proposed fine-tuning strategy is computationally efficient, requiring only a single GPU and approximately 6,500 training samples. Together with the authors’ stated intention to release the code upon acceptance, this supports reproducibility and facilitates adoption by the broader medical AI research community.

**Weaknesses:**

1.	Some aspects of the methodology and underlying design choices remain unclear or insufficiently justified. For instance, it is not specified which latent diffusion model (LDM) was used and the rationale for this choice. Similarly, the use of a CLIP model not pretrained on radiology-specific reports (as noted in Appendix A) requires clarification in the main text to support reproducibility and interpretability.
2.	The approach is heavily dependent on predictions from RadGraph-XL, which may introduce bias and limit the methodological scope. Relying on an external model constrains the generalizability of the approach and may amplify errors from RadGraph-XL in downstream tasks.
3.	The Related Work section does not fully account for relevant and recent developments. In Section 2.1, only methods up to 2023 are discussed, omitting newer approaches (e.g., [1], [2]), and Section 2.2 lacks sufficient detail on preceding work. The overall narrative is fragmented, making it difficult for readers to understand the rationale behind Section 2.3 and the identified scientific gap. Strengthening this section would clarify the novelty and positioning of the proposed method.
4.	While results include confidence intervals from bootstrapping, formal statistical significance analyses are missing. Furthermore, pathology-specific evaluations are limited. Distinctions between easily localized pathologies (e.g., pleural effusion) and smaller, dispersed pathologies are not systematically analyzed, which could better contextualize the localization performance of the proposed phase-grounding approach.
5.	The Gaussian lookup table (LUT) used for spatial grounding is relatively coarse, restricting localization granularity. Additionally, the vocabulary for anatomical localization is limited, potentially constraining the expressiveness of the model in fine-grained anatomical regions.

[1] Kumar, A., Kriz, A., Havaei, M., & Arbel, T. (2025). Prism: High-resolution & precise counterfactual medical image generation using language-guided stable diffusion. arXiv preprint arXiv:2503.00196.

[2] Huang, P., Gao, X., Huang, L., Jiao, J., Li, X., Wang, Y., & Guo, Y. (2024, May). Chest-diffusion: a light-weight text-to-image model for report-to-cxr generation. In 2024 IEEE International Symposium on Biomedical Imaging (ISBI) (pp. 1-5). IEEE.

**Detailed Comments:**

1.	The title could be revised to clearly indicate that the paper focuses on phrase grounding, improving immediate clarity for readers.
2.	Abbreviations should be introduced consistently at first mention and used uniformly thereafter. For example, “Latent Diffusion Models” is sometimes not introduced (see Abstract and Section 2.1), and abbreviations like OOD and CXR are used inconsistently. A thorough check and standardization of all abbreviations is recommended. Standardize abbreviations such as “SOTA” for state-of-the-art instead of “SoTA”.
3.	Avoid headings without text between them (e.g., Section 2 and 2.1, or 3 and 3.1). This structure should be corrected to improve readability.
4.	In Section 2.2, references should be added to support the statement “[…] most publicly available datasets […]”. In Section 3.2, the exact dataset used should be clearly named and cited.
5.	Formulas should be properly integrated into the text; currently, Equations (1), (2), (5), (6), and (7) are not consistently formatted.
6.	In formulas such as (1) and (2), it is recommended to use LaTeX brackets that fully enclose the content vertically to improve readability.
7.	In the supplementary material, table captions should appear above the tables, not below, to align with standard formatting conventions.
8.	Stylistically, a non-breaking space should be used before parentheses to avoid a parenthesis appearing at the beginning of a line after a line break, which improves text flow and readability.
9.	The last empty page can be removed for a cleaner layout, although this may be handled during final submission.

**Justification Of Final Rating:**

Based on the thorough rebuttal, the clarifications provided, and the planned improvements (including clearer methodological details, updated related work, and the release of code and pretrained models) I am confident in the quality and significance of the work and therefore raise my rating from 4 to 5 (strong accept).

**Justification Of The Preliminary Rating:**

The paper presents an interesting and promising approach with thorough experiments, including quantitative benchmarks, qualitative analyses, and downstream tasks. However, certain aspects of the methodology are not fully clear, such as the choice of the LDM, the use of a CLIP model not pretrained on radiology reports, and the reliance on RadGraph-XL. Some sections, including Related Work and localization evaluation, could also benefit from additional clarification and detail. I believe these issues can likely be addressed through the rebuttal and a careful revision, which is why I recommend a Weak Accept.

**Questions To Address In The Rebuttal:**

I would like the authors to provide explanations and clarifications regarding the weaknesses I have noted, including the methodological decisions behind some design choices. Additionally, I am interested in an assessment of how phase-grounding localization operates during inference—specifically, whether it is patient-specific or based on static precomputed representations.

---

> ### Author Response · Authors · 2026-01-25
> **Response to Reviewer xmN6**
>
> We appreciate your detailed feedback. Our point-by-point responses are provided below.
>
> > Some aspects of the methodology and underlying design choices remain unclear or insufficiently justified. For instance, it is not specified which latent diffusion model (LDM) was used and the rationale for this choice. Similarly, the use of a CLIP model not pretrained on radiology-specific reports (as noted in Appendix A) requires clarification in the main text to support reproducibility and interpretability.
>
> Thank you for your comment. As mentioned in Section 4.3 (pg. 9) and Appendix G (pg. 20), we use a publicly available LDM pre-trained on MIMIC-CXR. We also provide the link to download the pre-trained weights in Appendix G. Moreover, the LDM relies on the standard CLIP text encoder from the open-source Stable Diffusion 2.1 model, therefore it is not adapted to radiology reports.
>
> > The approach is heavily dependent on predictions from RadGraph-XL, which may introduce bias and limit the methodological scope...[abbreviated]
>
> Thank you for your comment. While we acknowledge the dependency on RadGraph-XL as a potential limitation (Appendix A), it also presents a clear opportunity to refine our proposed method in the future by incorporating more accurate clinical entity recognition models.
>
> > The Related Work section does not fully account for relevant and recent developments. In Section 2.1, only methods up to 2023 are discussed, omitting newer approaches (e.g., [1], [2]), and Section 2.2 lacks sufficient detail on preceding work...[abbreviated]
>
> Thank you for your suggestion. We have now updated the Related Work section to reflect recent advancements in chest X-ray LDMs, and to cite earlier works in medical phrase grounding.
>
> > While results include confidence intervals from bootstrapping, formal statistical significance analyses are missing. Furthermore, pathology-specific evaluations are limited. Distinctions between easily localized pathologies (e.g., pleural effusion) and smaller, dispersed pathologies are not systematically analyzed, which could better contextualize the localization performance of the proposed phase-grounding approach.
>
> Thank you for your suggestion. We have now included a discussion in Section 4.5.1 (pg. 10) of our revised manuscript.
>
> > The Gaussian lookup table (LUT) used for spatial grounding is relatively coarse, restricting localization granularity. Additionally, the vocabulary for anatomical localization is limited, potentially constraining the expressiveness of the model in fine-grained anatomical regions.
>
> Thank you for your comment. We would like to clarify that the Gaussian LUT serves as a coarse anatomical prior and our findings suggest that this level of granularity is sufficient to guide the LDM’s attention towards the region specified in the prompt. Regarding the vocabulary limitations, as noted in Appendix A, the list of anatomical locations used in this study is by no means exhaustive, and we identify its expansion as a promising direction for future research.
>
> > The title could be revised to clearly indicate that the paper focuses on phrase grounding, improving immediate clarity for readers.
>
> Thank you for your suggestion. While phrase grounding is a central evaluation task in our work, which is why it is featured as a primary keyword, the core contribution of the paper is a prompt tuning method that improves multi-modal alignment in text-to-image LDMs by optimizing the anatomy token embeddings. Moreover, we do not introduce a new heatmap extraction method for phrase grounding; instead, we rely on the approach introduced in [1] to extract such heatmaps. In addition to phrase grounding results, we provide evidence of the downstream utility of heatmaps through a grounded pneumothorax classification experiment (Appendix K), showing that improved localization translates to better classification performance. Therefore, we believe that the current title most accurately reflects the scope and contributions of our work.
>
> > Comments 2. - 9.
>
> Thank you for raising these points. We have revised the manuscript to incorporate the suggested edits.
>
> > Additionally, I am interested in an assessment of how phase-grounding localization operates during inference—specifically, whether it is patient-specific or based on static precomputed representations.
>
> Thank you for your comment. The phrase grounding pipeline used in this study operates on a per-image basis, where the input image is first inverted into noise using DDIM inversion and the final heatmap is extracted from intermediate cross-attention activations across layers and timesteps. As a result, since heatmaps are dynamically generated for each input during inference, localization is inherently patient-specific and does not rely on static precomputed representations.
>
>
> [1] Vilouras, Konstantinos, et al. "Zero-shot medical phrase grounding with off-the-shelf diffusion models." IEEE Journal of Biomedical and Health Informatics (2024).

---

> > ### Comment · Reviewer_xmN6 · 2026-01-29
> >
> > Thank you for the clarifications and detailed responses.
> >
> > Could you please specify exactly which statistical tests were performed? This should be clearly reported in the paper, and statistical significance could, for example, be indicated in the tables using asterisks.

---

> > > ### Author Response · Authors · 2026-01-31
> > >
> > > Thank you for your comment. Our conclusions in Section 4.5.1 (pg. 10) are based on the bootstrapped 95% confidence intervals reported in Table 1. We rely on non-overlapping confidence intervals as a conservative (i.e., sufficient but not necessary) criterion to identify statistically significant differences relative to the strongest baseline. While a paired hypothesis test could further complement this analysis, we note that non-overlapping intervals provide a robust indicator of significance. To improve clarity, we will add asterisks in Tables 1 and 2 of the final version to denote these significant differences.

---

### Official Review · Reviewer_bgfC · 2026-01-10

**Confidence:** 3
**Preliminary Rating:** 4
**Final Rating:** 4

**Summary:**

In this paper, the authors propose a new approach to improve the image-text alignment in VLP. They first curate a dataset by converting free-form radiology text into a standardized “{location} {pathology}” format, and further derive weak 2D supervision signals. They then use this data to fine-tune the token embeddings so that the model’s cross-attention is restricted to the specified anatomical locations. Evaluations on MS-CXR and VinDr-CXR datasets show that the proposed methods achieve good performance in the phrase grounding task.

**Strengths:**

(1) This paper addresses a practically important problem of attention leakage in LDM-based phrase grounding.

(2) The approach is parameter-efficient, as it fine-tunes only the token embeddings while keeping the main LDM components frozen.

(3) The authors also conduct comprehensive evaluations on both in-domain and out-of-distribution datasets, and include radiologist assessments of the generated heatmaps. Overall, the results support the effectiveness of the proposed approach.

(4) The manuscript is well written, and the method is clearly explained.

**Weaknesses:**

(1) Since the curated dataset contains a predefined set of 27 anatomical locations and 8 pathologies, with a simple format of  “{location} {pathology}”, it may limit the capability of the finetuned model in phase grounding and bias it towards the predefined locations.

(2) As the 2D Gaussian parameters are derived from annotations in Chest Imagenome, the anatomical location distributions may be biased by this dataset, thereby limiting the generalizability of the proposed method.

(3) As discussed in the appendix, RadGraph-XL exhibits several failure modes when extracting anatomical locations. Since the curated dataset is central to the fine-tuning process, I wonder whether the authors considered or experimented with more advanced general-purpose models (e.g., GPT) for the data curation step.

**Detailed Comments:**

Although the paper discusses heatmaps in several places, the procedure for generating heatmaps for the proposed model and the compared baselines is not entirely clear. Since the heatmap is central to all evaluations, I suggest adding a dedicated section that explicitly describes (1) how the heatmap is computed from the cross-attention layers (e.g., which layers/timesteps are used and how they are aggregated), and (2) how the heatmap is normalized and thresholded for evaluation. A self-contained, detailed description would significantly improve clarity and reproducibility.

**Justification Of Final Rating:**

In their rebuttal, the authors provided additional clarification and expanded discussion addressing my major concerns. Overall, I find the work solid and well-executed, and the responses have strengthened my confidence in the paper’s contributions. I therefore recommend the paper for acceptance.

**Justification Of The Preliminary Rating:**

This work addresses an important issue of attention leakage in LDM-based phrase grounding with a parameter-efficient finetuning approach. The proposed method shows strong performance in the evaluations. However, the method may be constrained by the curated prompt space (27 locations/8 pathologies) and by potential biases from Chest ImaGenome-derived Gaussian priors.

**Questions To Address In The Rebuttal:**

I suggest that the authors include a more detailed discussion on the weakness mentioned above.

---

> ### Author Response · Authors · 2026-01-25
> **Response to Reviewer bgfC**
>
> We appreciate the reviewer’s constructive feedback. Please find our detailed responses to each of the points raised below:
>
> > Since the curated dataset contains a predefined set of 27 anatomical locations and 8 pathologies, with a simple format of “{location} {pathology}”, it may limit the capability of the finetuned model in phase grounding and bias it towards the predefined locations.
>
> Thank you for your comment. While our study focuses on 8 pathologies, which are dictated by the available labels in the MS-CXR dataset, those categories cover a wide range of appearances and locations, suggesting that our findings are generalizable to other chest-Xray pathologies. Regarding the role of the 27 anatomical locations, we would like to clarify that these are purely used as a methodological tool and do not influence the evaluation, since the ground truth bounding boxes can be located anywhere in the image.
>
> > As the 2D Gaussian parameters are derived from annotations in Chest Imagenome, the anatomical location distributions may be biased by this dataset, thereby limiting the generalizability of the proposed method.
>
> Thank you for raising this point. We agree that the 2D Gaussian parameters derived from Chest Imagenome annotations reflect the location distributions present in that dataset. To assess whether this introduces a dataset-specific bias that might limit generalizability, we evaluated the model on the out-of-distribution (OOD) dataset VinDr-CXR (Table 2, pg. 11). This dataset differs significantly from the pre-training data (MIMIC-CXR) in terms of the patient population, imaging characteristics, and the ground truth pathology bounding box statistics. Despite these shifts, our model achieves strong performance on VinDr-CXR, indicating that the proposed method is robust to such variations.
>
> > As discussed in the appendix, RadGraph-XL exhibits several failure modes when extracting anatomical locations. Since the curated dataset is central to the fine-tuning process, I wonder whether the authors considered or experimented with more advanced general-purpose models (e.g., GPT) for the data curation step.
>
> Thank you for this comment. Although we acknowledge that RadGraph-XL [1] exhibits failure modes, our decision to use it was based on existing benchmarks comparing specialized clinical entity recognition models to general-purpose LLMs. Specifically, the authors of RadGraph-XL evaluated GPT-4 on the same task, and their results (shown in Table 12 in [1]) show that their specialized model significantly outperformed GPT-4. Given this performance gap, we concluded that the domain-specific RadGraph-XL model, despite its limitations, remains more reliable than current general-purpose models for curating anatomical locations.
>
> > Although the paper discusses heatmaps in several places, the procedure for generating heatmaps for the proposed model and the compared baselines is not entirely clear. Since the heatmap is central to all evaluations, I suggest adding a dedicated section that explicitly describes (1) how the heatmap is computed from the cross-attention layers (e.g., which layers/timesteps are used and how they are aggregated), and (2) how the heatmap is normalized and thresholded for evaluation. A self-contained, detailed description would significantly improve clarity and reproducibility.
>
> Thank you for your suggestion. We have now updated our manuscript to include a detailed description of the heatmap extraction process for each baseline model used in our study, including ours (please refer to Section 4.3, pg. 9).
>
> [1] Delbrouck, Jean-Benoit, et al. "Radgraph-xl: A large-scale expert-annotated dataset for entity and relation extraction from radiology reports." Findings of the Association for Computational Linguistics: ACL 2024. 2024.

---

### Official Review · Reviewer_rZuf · 2026-01-10

**Confidence:** 4
**Preliminary Rating:** 5
**Final Rating:** 5

**Summary:**

This paper presents a method to improve image–text alignment in biomedical vision–language pretraining by deriving pathology localization supervision directly from unstructured radiology reports. The approach combines clinical entity recognition with anatomical region annotations to generate weak localization cues, and introduces an efficient fine-tuning strategy that steers cross-attention in a pre-trained latent diffusion model toward anatomically specified regions. The method is evaluated on the MS-CXR phrase grounding benchmark and on an out-of-distribution dataset (VinDr-CXR).

**Strengths:**

Extensive experiments were performed to support the methodical contributions. Solves two significant issues in multi-modal medical image analysis-- (i) a weakly supervised mechanism to generate ground truth required for supervision, and (ii) solves the task of phrase grounding utilizing the constraints that focus on aligning the text and image feature representations.

**Weaknesses:**

Generally, the paper is well-structured and includes adequate validation, but there can be a concept diagram that can clearly explain how and which layer of the LDM attention contributes to the final bounding box extraction. This can help the reader explicitly understand the task, the usage of LDM with the loss contributions of the paper.

**Detailed Comments:**

1. Inclusion of a more explicit concept diagram to get the entire task and the contributions clearly.
2. More visualizations from the OOD dataset can help assess the generalizations.
3. Visual images showing how the 'trg' appears would help in understanding.
4. The interplay of both the loss components can be analyzed with varying ratios.
5. Is there a role of the images generated by the LDM in the final attention extraction?

**Justification Of Final Rating:**

I thank the authors for the clarifications.

All the concerns are well addressed; specifically, the addition of the overall concept diagram makes the readability better. So I recommend acceptance of the paper.

**Justification Of The Preliminary Rating:**

A strong proposition with sufficient experimental validation for the multimodal analysis. The methodical contribution is new and helps in model interpretability. This contribution sets a good base for better text and image feature alignment.

**Questions To Address In The Rebuttal:**

1. Certain decoding layers were used for constraining the attention with the losses. Can the authors provide more details regarding why these specific layers were chosen? Will this affect the attention of other layers?
2. Can the authors provide more details regarding-- Will this affect the image generation quality of the LDM?
3. Also, will the image generation capability be connected to the attention learnt by the model?
4. Can the authors provide more details regarding the interplay of both the loss components with varying ratios?

---

> ### Author Response · Authors · 2026-01-25
> **Response to Reviewer rZuf**
>
> We appreciate your positive feedback on our work. Based on your comments, we have now revised the manuscript to include:
> - A concept diagram that provides a high-level description of the task and our contributions (in pg. 7). This diagram also shows an example of a ‘trg’ image that is used as a supervision signal during fine-tuning. [Comment 1. and 3.]
> - Visualizations from the OOD dataset VinDr-CXR (in the Appendix, pg. 28). [Comment 2.]
> - An ablation study on the weighting coefficients of each loss term (in the Appendix, pg. 23). [Comment 4]
>
> Our responses to your questions are provided below.
> > Is there a role of the images generated by the LDM in the final attention extraction?
>
> Thank you for your question. The generated images do not play a role in the attention extraction process. Instead, following [1], we invert the original image into noise using DDIM inversion and extract cross-attention feature maps from intermediate layers and diffusion timesteps.
>
> > Certain decoding layers were used for constraining the attention with the losses. Can the authors provide more details regarding why these specific layers were chosen? Will this affect the attention of other layers?
>
> Thank you for your comment. Following prior works that repurpose text-to-image diffusion models for tasks such as image editing [2] and keypoint detection [3], we focus on cross-attention layers in the U-Net's decoder since those yield more semantically rich features. Moreover, although we do not modify other attention layers, the effects of fine-tuning naturally propagate to subsequent denoising timesteps.
>
> > Can the authors provide more details regarding-- Will this affect the image generation quality of the LDM?
>
> Thank you for your comment. We would like to clarify that our design choice guides the model towards learning task-specific features rather than preserving its original image synthesis capabilities; as a result, prompt tuning does affect image generation quality. Moreover, in cases where downstream performance is not the primary objective, the original text embeddings used during LDM pre-training can be retained to enable high-quality text-to-image synthesis.
>
> > Also, will the image generation capability be connected to the attention learnt by the model?
>
> Thank you for your comment. Cross-attention is a core mechanism of the LDM since it directly integrates information derived from text into visual content. As a result, the attention patterns learned by the model during pre-training directly influence its generative capabilities.
>
> > Can the authors provide more details regarding the interplay of both the loss components with varying ratios?
>
> We appreciate your suggestion. We have now added an ablation study to the revised manuscript (pg. 23) showing the effect of each loss weighting coefficient on downstream performance. Overall, we observed an average performance drop on MS-CXR-loc of 0.15 and 0.21 in terms of CNR, and a consistent 10\% decrease in terms of mIoU when we vary those coefficients.
>
> [1] Vilouras, Konstantinos, et al. "Zero-shot medical phrase grounding with off-the-shelf diffusion models." IEEE Journal of Biomedical and Health Informatics (2024).
>
> [2] Hertz, Amir, et al. "Prompt-to-Prompt Image Editing with Cross-Attention Control." The Eleventh International Conference on Learning Representations.
>
> [3] Hedlin, Eric, et al. "Unsupervised keypoints from pretrained diffusion models." Proceedings of the IEEE/CVF Conference on Computer Vision and Pattern Recognition. 2024.

---

### Author Rebuttal · Authors · 2026-01-25

**Rebuttal:**

We appreciate the reviewers’ feedback on our submitted work. In the revised manuscript, we have now added the following (note that all changes are highlighted in red in the revised manuscript):

- An introductory paragraph for Section 2 “Related Work” (pg. 3) and Section 4 “Experiments” (pg. 8). [reviewer xmN6]

- Citations of recent literature on chest X-ray text-to-image Latent Diffusion Models and earlier approaches in medical phrase grounding to provide a more comprehensive overview of the field in Section 2 “Related Work” (pg. 3-4). [reviewer xmN6]

- A concept diagram in Section 3.2 (pg. 7) that provides an overview of our proposed prompt tuning method. [reviewer rZuf]

- A dedicated Section 4.3 “Baseline models and heatmap extraction” (pg. 9) that introduces all baselines, including our fine-tuned LDM, and the methodology used to extract heatmaps from each model to ensure transparency in our experimental setup. [reviewer bgfC]

- A discussion in Section 4.5.1 (pg. 10) regarding the reliability of our results, focusing on non-overlapping (or minimally overlapping) confidence intervals between our model and the strongest baseline to further support the robustness of the observed performance gains. [reviewer xmN6]

- An ablation study on the weighting coefficients of each loss term in Appendix H.2 (pg. 23) to further justify our design choice. [reviewer rZuf]

- Heatmap visualizations from the OOD dataset VinDr-CXR in Appendix L (pg. 28). [reviewer rZuf]

- Minor formatting edits throughout the text. [reviewer xmN6]

**Supporting Material:**

/attachment/2f5d9abcbd49b13be7f606c215e3a004ec30cf62.pdf

---

### Meta-Review · Area_Chair_a9wS · 2026-02-02

**Recommendation:** Accept (Oral)
**Confidence:** 4

**Metareview:**

All three reviewers found this paper to be a strong contribution to the field. The main points raised during review were largely focused on clarity, completeness, and justification of design choices rather than fundamental concerns about correctness or significance. The authors responded to all reviewers. In the rebuttal+revision, the authros added a concept diagram and additional qualitative results (rZuf), introduced a dedicated section detailing baselines and heatmap extraction procedures (bgfC), expanded and updated the related work, clarified model and dataset choices, and improved methodological transparency and formatting (xmN6). The authors also provided additional ablations, OOD evaluations on VinDr-CXR, and discussions of robustness and limitations, addressing concerns about generalization and bias raised by multiple reviewers. While some limitations remain, such as reliance on RadGraph-XL for weak supervision and a constrained anatomical vocabulary (bgfC, xmN6), these are well acknowledged by the authors and framed as reasonable trade-offs rather than flaws that undermine the contribution. Overall the AC believes this work will intrigue valuable discussions at MIDL.

---

### Decision · Program_Chairs · 2026-02-13

Accept (Poster)